# Double-Edged Lipid Nanoparticles Combining Liposome-Bound TRAIL and Encapsulated Doxorubicin Showing an Extraordinary Synergistic Pro-Apoptotic Potential

**DOI:** 10.3390/cancers11121948

**Published:** 2019-12-05

**Authors:** Diego De Miguel, Ana Gallego-Lleyda, Miguel Martinez-Ara, Javier Plou, Alberto Anel, Luis Martinez-Lostao

**Affiliations:** 1Departamento de Bioquímica, Biología Molecular y Celular, Universidad de Zaragoza, 50009 Zaragoza, Spain; annaiss89@hotmail.com (A.G.-L.); mmartinezara@gmail.com (M.M.-A.); javierplou@gmail.com (J.P.); anel@unizar.es (A.A.); 2Cell Death, Cancer and Inflammation Group, University College of London, London WC1E 6BT, UK; 3Instituto de Investigación Sanitaria de Aragón (ISS), 50009 Zaragoza, Spain; 4Servicio de Inmunología, Hospital Clínico Universitario Lozano Blesa, 50009 Zaragoza, Spain; 5Departamento de Microbiología, Medicina Preventiva y Salud Pública, Universidad de Zaragoza, 50009 Zaragoza, Spain; 6Instituto de Nanociencia de Aragón, 50018 Zaragoza, Spain; 7Instituto de Ciencias de los Materiales de Aragón, 50009 Zaragoza, Spain

**Keywords:** TRAIL, liposomes, cancer immunotherapy, caspase-8, doxorubicin, endocytosis

## Abstract

Although TRAIL (TNF-related apoptosis-inducing ligand, also known as Apo2L) was described as capable of inducing apoptosis in transformed cells while sparing normal cells, limited results obtained in clinical trials has limited its use as an anti-tumor agent. Consequently, novel TRAIL formulations with enhanced bioactivity are necessary for overcoming resistance to conventional soluble TRAIL (sTRAIL) exhibited by many primary tumors. Our group has generated artificial liposomes with sTRAIL anchored on their surface (large unilamellar vesicle (LUV)-TRAIL), which have shown a greater cytotoxic activity both in vitro and in vivo when compared to sTRAIL against distinct hematologic and epithelial carcinoma cells. In this study, we have improved LUV-TRAIL by loading doxorubicin (DOX) in its liposomal lumen (LUVDOX-TRAIL) in order to improve their cytotoxic potential. LUVDOX-TRAIL killed not only to a higher extent, but also with a much faster kinetic than LUV-TRAIL. In addition, the concerted action of the liposomal DOX and TRAIL was specific of the liposomal DOX and was not observed when with soluble DOX. The cytotoxicity induced by LUVDOX-TRAIL was proven to rely on two processes due to different molecular mechanisms: a dynamin-mediated internalization of the doxorubicin-loaded particle, and the strong activation of caspase-8 exerted by the liposomal TRAIL. Finally, greater cytotoxic activity of LUVDOX-TRAIL was also observed in vivo in a tumor xenograft model. Therefore, we developed a novel double-edged nanoparticle combining the cytotoxic potential of DOX and TRAIL, showing an exceptional and remarkable synergistic effect between both agents.

## 1. Introduction

TRAIL (TNF-related apoptosis-inducing ligand, also known as Apo2L) is a death ligand that belongs to the TNF cytokine super-family. Its selective pro-apoptotic ability on transformed cells, leaving normal cells unaffected [1,2,3], sparked the interest in using this protein as a potential therapeutic strategy in cancer [4,5]. In fact, a large number of clinical trials were carried out using TRAIL-based therapy on a wide variety of human cancers, reaching phase II/III [6]. Unfortunately, although proven safe, TRAIL-based therapies showed very limited therapeutic activity in clinical trials [7,8,9]. Therefore, better sensitization strategies can be useful in order to overcome the resistance of cancer cells to TRAIL [10,11,12,13]. In this line, a wide variety of drugs of different nature that can act in combination with TRAIL, sensitizing cancer cells to apoptosis induced by this death ligand, have been described. In fact, TRAIL has been combined, among others, with the proteasome inhibitor bortezomib [14], the cdk inhibitor flavopiridol [15], the histone deacetylase inhibitor vorinostat [16], the flavonoid quercetin [17], and the cytostatic drugs paclitaxel [18], actinomycin D [19], or doxorubicin [20]. On the other hand, novel TRAIL formulations with improved bioactivity can also be of great usefulness for its future clinical application in order to overcome TRAIL resistance [21,22,23,24,25,26]. These TRAIL formulations include highly stable TRAIL trimers [3,27], fusion proteins with single-chain variable antibody fragments (scFv) that active target TRAIL to cancer cells [28,29,30,31] or immune cells [32,33], mutant variants with specificity to DR5 [34,35], and different types of nanoparticles [36,37,38,39,40,41,42]. In this line, our group generated an original formulation of TRAIL based on immobilizing TRAIL molecules on the surface of artificial lipid nanoparticles (large unilamellar vesicle (LUV)-TRAIL), resembling the natural TRAIL-containing exosomes secreted by activated human T-cells [43,44]. We already demonstrated that LUV-TRAIL exhibited an enhanced anti-tumor activity when compared to soluble TRAIL (sTRAIL) against leukemic cells resistant to TRAIL or to chemotherapeutic drugs [37,45,46], and on epithelial carcinoma cells both in vitro and in vivo [47,48,49]. The increased cytotoxicity of LUV-TRAIL relied on its ability to induce an improved DR5 clustering leading to an enhanced DISC recruitment in comparison to sTRAIL [45,49]. However, although LUV-TRAIL has proven more efficient than sTRAIL in all tumor cell lines tested, some cancer cell lines remain somewhat resistant to its pro-apoptotic activity. Therefore, seeking to optimize the original LUV-TRAIL formulation to overcome the resilient TRAIL-resistance of some cancer cells, we have generated a novel liposomal formulation combining LUV-TRAIL with a chemotherapeutic drug encapsulated inside the lipid nanoparticle. Concerning the drug to be encapsulated, after performing a research work and considering several options among the already described liposomal drugs, liposomal doxorubicin (DOX) was finally chosen. Liposomal DOX was one of the first encapsulated drugs to be developed, more than 20 years ago, and its anti-tumor bioactivity has been assessed in a great number of scientific articles (reviewed in Reference [50]). In this line, the use of liposomal DOX assembled to the erythrocyte leveraged chemotherapy (ELeCt) platform in cancer treatment has been recently reported [51]. In this work, the authors demonstrate that liposomal DOX using the ELeCt platform is highly effective against lung metastasis in a melanoma model. Doxorubicin has been described to synergize with TRAIL when administered in combination in several studies [38,52,53,54,55]. Beyond DOX has been reported to synergize with TRAIL, its liposomal was chosen due to scientific literature about the preparation and characterization of liposomal DOX, which can be easily found (reviewed in Reference [56]). Moreover, from an experimental point of view, the encapsulation protocol of DOX perfectly suited the coupling process of TRAIL with LUV, so both processes (encapsulation and coupling) could be easily combined with minor modifications. Thus, we devised the generation of a unique formulation combining both TRAIL and DOX (LUVDOX-TRAIL), which could be of great interest to overcome the resistance of different types of cancer. Therefore, we used the liposomal particle of LUV-TRAIL as a vehicle for DOX, resembling the well-known liposomal formulation of DOX, but combining it with the superior pro-apoptotic potential of LUV-TRAIL, generating LUVDOX-TRAIL. This double-edged new formulation can potently and simultaneously trigger the intrinsic and the extrinsic apoptotic pathways to induce a great cytotoxic effect against tumor cells. Indeed, this new liposomal formulation exhibited an outstanding pro-apoptotic potential, greatly improving the performance of LUV-TRAIL in all cancer cells tested. It is noteworthy that LUVDOX-TRAIL was capable of overcoming the resistance to LUV-TRAIL in highly TRAIL-resistant cancer lines. Moreover, LUVDOX-TRAIL with a defined low concentration of DOX maintained their greater anti-tumor effect both in vitro and in vivo while sparing normal cells. Our results show that this optimized double-edged LUV-TRAIL formulation harbors an enhanced exceptional cytotoxic ability and could be potentially useful to improve anti-tumor therapy against TRAIL-resistant cancers.

## 2. Results

### 2.1. Synthesis and Characterization of Large Unilamellar Vesicle Doxorubicin-TNF-Related Apoptosis-Inducing Ligand (LUVDOX-TRAIL)

Large unilamellar vesicles (LUVs) encapsulating DOX inside their lumen and decorated with TRAIL on their surface (LUVDOX-TRAIL (LDT)) (Figure 1a) were synthetized as outlined in the protocol shown in Figure 1b. DOX was encapsulated inside the liposomal lumen using the pH gradient method, as described in the Material and Methods Section. In short, this method relies on the generation of a pH difference between the liposomal lumen (low pH) and the external media (neutral pH). Thus, DOX molecules passively diffuse from the media to the liposomal lumen, where the low pH induces a protonation of the DOX molecule, rendering it insoluble and therefore unable to cross the lipid barrier, being entrapped inside the liposomes. As shown in Figure 1d, this method rendered an entrapment efficiency of around 97%. Assuming this encapsulation efficiency, the final concentration of encapsulated DOX was 1.4 mM (Figure 1c). In contrast, as expected, classic LUVs bearing PBS in the liposomal lumen (LUV-PBS) showed a rather poor encapsulation efficiency. Cryo-TEM assessment confirmed the presence of a homogeneous population of liposomal particles of around 100 nm of diameter, similarly to LUV-TRAIL (LT) [57]. Of note, a black aggregate inside the liposomes could also be observed, corresponding to the precipitated insoluble DOX, which is a typical feature of liposomal DOX encapsulation (Figure 1e).

Once the feasibility and efficiency of the DOX encapsulation method were proven, the next step was to attach sTRAIL on their surface. For that, sTRAIL was incubated with LUVDOX (LD) at the same protein/lipid ratio as in the classic LT [37] to obtain the LDT formulation. Importantly, sTRAIL was coupled to LD by incubating at 15 °C instead of the classic 37 °C used for LT to diminish possible leakage of DOX. However, this lower incubation temperature did not compromise sTRAIL attachment to the liposomal surface. As shown in Figure 1f, the coupling efficiency was very high and similar to that of classic LT [57], validating the modification of the protocol. Therefore, the final concentration of DOX and TRAIL within LDT was 775 μM and 12 μg/mL, respectively.

### 2.2. In Vitro Cytotoxic Potential of LUVDOX-TRAIL

The anti-tumor activity of LDT was tested on a panel of cell lines, composed mainly by TRAIL-resistant cell lines (Appendix A). As shown in Figure 2, LDT showed an exceptional cytotoxic activity on all cell lines tested, proving more efficiency than LT and LD, respectively. In this regard, although LD alone induced some cytotoxicity on some cell lines, this was rather limited in most cases (Figure 2a). The cells presented the typical features of apoptotic cell death, such as blebbing and cell shrinkage. Importantly, the effect of LDT was not observed when LT was co-administered in combination with increasing concentrations of soluble DOX up to more than 15-fold that of LDT (Figure 2c), suggesting a specific synergism between the liposomal DOX and liposome-bound TRAIL. Next, we measured the killing kinetic of LDT. As shown in Figure 2d, LDT showed a much faster cytotoxic kinetic than both LD and LT. In all epithelial cancer cell lines, LDT reached the maximum cell death after six hours (Figure 2d, upper panels). Moreover, in A549 and SKBR3 cells, the onset of cell death could be observed already after two hours. On the other hand, LDT showed a slower kinetic on sarcoma cell lines, reaching maximum levels at 20 hours (Figure 2d, bottom panels). In all cases, LD only showed a limited cytotoxic exposure effect, with LDT being significantly more potent in all cases. Altogether, our results show a unique synergistic effect between the surface-bound TRAIL and the encapsulated DOX, which cannot be achieved by LT, LD, or an equivalent concentration of soluble DOX.

### 2.3. LUVDOX-TRAIL are able to Induce a Stronger Activation of the Extrinsic Apoptotic Pathway than LUV-TRAIL in Cancer Cells

Next, we set out to characterize the nature of the cell death induced by LDT. First, the role of TRAIL in the cytotoxicity exerted by LDT was analyzed by blocking TRAIL (Figure 3a). Pre-incubation with the TRAIL-blocking antibody RIK completely protected all cell lines from LDT-induced cytotoxicity. On the other hand, the exposure of phosphatidylserine detected by annexin-V staining in the cytotoxicity experiments suggested a classic apoptotic process. To corroborate that, the role of caspases in LDT was explored. First, sarcoma cell lines HT-1080 and RD were incubated with sTRAIL, LT, LD, and LDT for 20 hours and activation of the main caspases involved in the extrinsic apoptotic pathway was assessed by Western blot (Figure 3b, upper panels). Activation of caspase-8 and caspase-3 was clearly increased when sarcoma cells were treated with LT compared to sTRAIL, as previously described [58]. Moreover, cleavage of Bid and PARP-1, the specific substrates for caspases-8 and -3 respectively, correlated with the activation of both caspases. It is noteworthy that LD had no effect on caspase activation. In contrast, LDT induced a stronger caspase activation than both LD and LT, which correlated with a higher cell death induction in the same experiments (Figure 3b, bottom panels). When analyzed in a time-course setting, LDT again showed a much faster ability to activate caspases -8 and -3 (Figure 3c). It is worth noting that LDT also induced a quick and strong activation of caspase-9. Overall, while LD did not induce any noticeable activation of any of the three caspases analyzed, LDT induced a clear and strong activation of the three caspases even from the 30-minute time point. Interestingly, the three caspases seemed to be activated simultaneously. With that aim, analysis of caspase activation after pre-incubation with the pan-caspase inhibitor z-VAD-fmk was carried out in sarcoma cells (Figure 3d). Treatment with z-VAD-fmk abrogated caspase activation almost completely both in HT-1080 and RD cells treated with LT and with LDT. Moreover, cleavage of Bid and PARP-1, the specific substrates for caspases-8 and -3 respectively, were also fully inhibited when cells were treated with z-VAD-fmk. Having corroborated that LDT induced a strong caspase activation, we next checked if caspases were the main driver of LDT cytotoxicity. Thus, A549, SKBR3, and HT-29 cells were subjected to LDT treatment, in the presence or absence of the pan-caspase inhibitor z-VAD-fmk or the specific caspase-8 inhibitor z-IETD-fmk (Figure 3e). Both caspase inhibitors were able to revert cell death almost completely. All the results pointed towards LDT-induced apoptosis being a purely caspase-8-dependent process. In order to corroborate the role of caspase-8, the rhabdomyosarcoma cell line RH4, which is known to lack caspase-8 [59], was also studied. LDT did not have any cytotoxic effect on RH4 cells (Appendix A), therefore corroborating the main role ofcaspase-8 in LDT-induced apoptosis. In this line, silencing caspase-8 in SKBR3 cells completely abrogated LDT-induced cell death (Appendix A). Finally, given the double-edged nature of LDT, being a priori able to trigger both the extrinsic and the intrinsic apoptotic pathways, and considering the strong activation of caspase-9 observed in Figure 3c, we also checked the activation of the intrinsic apoptotic pathway by using modified HCT-116 cells lacking both Bax and Bak (HCT-116 BB), therefore lacking a functional mitochondrial apoptotic pathway [49] (Appendix A). HTC-116 BB cells were completely resistant to LDT, suggesting an important role for the mitochondrial apoptotic pathway in LDT-induced cell death.

### 2.4. LUVDOX-TRAIL-Induced Cytotoxicity is a Dynamin-Dependent Mechanism Involving Caspase-8 Activity

Although the involvement of TRAIL in LDT cytotoxicity was clear, the contribution of DOX to the cell death was still elusive. Indeed, the uptake mechanism of LD and LDT was still unclear. To perform a more detailed analysis of the role of DOX in both liposomal formulations (LD and LDT), A549 cells were subjected to time-course stimulation with both liposomal formulations, and DOX fluorescence was analyzed by microscopy. As shown in Figure 4a, both LD and LDT could be readily detected inside the cells already after 15 minutes of treatment. This fluorescence remained mostly unchanged for LD, whereas in LDT-treated cells, the red fluorescence pattern dramatically changed, adopting a very condensed aspect over time, and localizing in what could be identified as the nucleus. It is worth noting that no apoptosis could be detected in LD-treated cells, correlating with our previous results. In addition, the sharp DOX condensation only occurred in cells undergoing apoptosis (white arrows in T = 120), suggesting a cause–effect correlation between apoptosis and DOX localization in the nucleus.

Interestingly, the fluorescence pattern observed for both LD and LDT at short times appeared to localize in a somehow defined cellular compartment, rather than being dispersed in the cytosol (T = 15). Having proven that encapsulated DOX both in LD and LDT was endocytosed, and taking into consideration previous studies reporting the lysosomal localization of DOX after treatment with liposomal doxorubicin [60], we considered the possibility of DOX being localized in the endosomal compartment. To corroborate this, a specific lysosomal marker (Lysotracker™) was used in parallel to LD or LDT administration (Figure 4b). After 60 minutes of exposure, almost all DOX fluorescence co-localized with Lysotracker™ in A549 cells treated with LD. However, when treated with LDT, most of DOX fluorescence already seemed to be located at the nucleus, whilst only a faint red fluorescence could be observed outside the nucleus, seemingly co-localizing with the lysosomal tracker, suggesting that at least part of the DOX would still remain in the lysosomes.

Next, we characterized the involvement of the endocytic pathway in LD- and/or LDT-induced cell death using the specific dynamin inhibitor Dynasore (DYN) [61]. A549 and SKBR3 cells were pre-treated with DYN for one hour prior to the addition of LD or LDT and apoptosis induction was measured at three hours to analyze the early cell death induced by LDT (Figure 4c). Of note, DOX uptake by the cells was also assessed by measuring its characteristic red fluorescence by flow cytometry. In the absence of DYN, cytotoxicity perfectly correlated with previous data. When cells were pre-treated with DYN, cell death induced by LDT was abrogated in both cell lines. Regarding DOX fluorescence, in general, both LD and LDT presented high MFI levels in both cell lines, although fluorescence was slightly higher for LDT. Interestingly, similarly to apoptosis levels, fluorescence levels were significantly reduced by DYN. All of this data strongly suggested a role for dynamin in the uptake of the liposomal DOX into the lysosomal compartment, regardless of the presence of TRAIL on the liposomal surface and proved that DOX requires to be internalized in order to exert its cytotoxicity. A deeper analysis by confocal microscopy in A549 cells following a time-course treatment with either LD or LDT further corroborated these observations (Figure 4d). Thus, while DOX could be readily detected colocalizing with the lysosomal marker LAMP-1 already after 5 minutes of stimulation with LDT, DOX could only be clearly detected after 30 minutes of stimulation in the case of LD. Moreover, in LDT-treated cells, DOX fluorescence co-localized with DAPI already after 15 minutes, and almost all DOX was present inside the nuclei after 30 minutes. In the case of LD, the uptake was much slower, with DOX only beginning to be clearly detected inside the cells after 30 minutes, and overall to a lesser extent than LDT.

Remarkably, when A549 cells were pre-treated with DYN, co-localization of DOX with DAPI was dramatically decreased both in LD-treated and LDT-treated cells. Interestingly, pre-treatment with z-VAD-fmk also decreased the amount of DOX into the nuclei, although to a lesser extent than DYN. When analyzed by Western blot, LT and LDT showed a clear activation of the caspase cascade, although LDT again showed a more robust activation and cleavage of all caspases and substrates analyzed (Figure 4e). LD did not induce any activation of any caspase, including caspase-9. As expected, caspase activation was reduced after z-VAD-fmk treatment. Interestingly, pre-treatment with DYN also diminished caspase activation.

Finally, we wanted to assess whether LT administered in combination with LD should render a similar outcome to LDT. For that, experiments using LD along with LT were carried out on A549 and SKBR3 cells, comparing with LDT. To dissect the roles of the endocytic process on the one hand and of TRAIL-induced caspase-8 activation on the other, cells were pre-treated with DYN and with z-IETD-fmk, and both apoptosis induction and DOX fluorescence were measured (Figure 4f). LDT was again the most cytotoxic agent. Interestingly, the combination of LD and LT showed a substantial synergy to induce cell death in both cell lines. Importantly, both DYN and z-IETD-fmk were able to abrogate apoptosis induced by LDT or by the combination of LT and LD. Regarding intracellular fluorescence, the result was similar to that observed in Figure 4a, where LD and LDT showed similar fluorescence patterns. In this case, the combination of LD and LT rendered similar results to LD alone or LDT. Moreover, the dynamin inhibitor DYN not only blocked DOX fluorescence upon treatment with LD or LDT, but also with the combination of LT and LD. Remarkably, the caspase-8 inhibitor z-IETD-fmk significantly reduced DOX fluorescence levels in the points where TRAIL was involved, indicating that DOX fluorescence intensity was also dependent of caspase-8 activity.

### 2.5. Reduction of DOX Concentration in LUVDOX-TRAIL Reduces Their Toxicity on Normal Peripheral Blood Mononuclear Cells (PBMC) but Maintain Cytotoxicity on Cancer Cells

After assessing the greater cytotoxic effect of LDT in all cancer cell lines tested, and having characterized its mechanism of action, we next sought to test the possible cytotoxicity of this new formulation on normal primary cells. For that purpose, cytotoxicity of LDT was analyzed on freshly isolated peripheral blood mononuclear cells (PBMC) and on activated T-cells (or T-cell blasts) generated from those PBMC (Appendix A). LDT induced cell death on PBMC and on day-6 T-cell blasts after 24 hours of treatment when used at the usual concentrations of DOX and TRAIL (64 μM of DOX and 1 μg/mL of TRAIL, respectively) (Appendix A). Toxicity was similar on both CD4^+^ and CD8^+^ 6-day T-cell blasts subpopulations (Appendix A), and it was clearly due to DOX, since LD exhibited a similar cytotoxicity pattern. In order to decrease LDT toxicity on normal cells, LDT with a concentration of DOX 10 times lower than those previously described (hereafter LDT 1/10) but with the same concentration of TRAIL were generated. Reduction of the concentration of encapsulated DOX to 6.4 μM eliminated the toxicity of LDT on freshly isolated PBMCs and considerably reduced the cytotoxicity on 6-day T-cell blasts (Appendix A). Finally, the cytotoxic potential of LDT 1/10 was assessed, as in Figure 2, using the same cell lines (Figure 5). Reduction of encapsulated DOX in liposomal composition reduced the cytotoxic potential of LD and LDT. However, although slightly reduced, importantly, LDT 1/10 still retained a strong activity against all cancer lines tested, still being the most potent formulation (Figure 5).

### 2.6. In Vivo Cytotoxic Potential of LUVDOX-TRAIL

Next, we sought to assess the in vivo efficiency of LDT. However, and given the high cytotoxicity of LDT in vitro, we first checked the safety of LDT, to rule out any overt adverse effect in vivo. Thus, both LT and LDT were systemically injected in mice. No sign of inflammatory infiltrate, fibrosis, or necrosis in any tissue were observed (Figure 6a). Moreover, there was no increase in ALT or AST serum levels after treatment with LDT, excluding in vivo hepatotoxicity, a well-known adverse effect described for TRAIL (Figure 6b) [62]. Therefore, we could conclude that LDT do not present systemic toxicity, and they can be used as a therapeutic agent in vivo.

Next, we tested the efficacy of LDT on a subcutaneous xenograft model. Thus, Hsd:Athymic Nude-Foxn1^nu^ mice bearing subcutaneous HT-1080 tumors were treated intraperitoneally with daily injections of either LT or LDT 1/10 for four consecutive days (described in the Materials and Methods Section). Tumor volumes were monitored, and mice were culled when the tumor volume reached 100 mm^2^. All control mice were culled at day 13, while mice treated with LT survived until day 18. In the case of mice treated with LDT, 66.7% (4 animals of 6 included in this group) survived until day 20 (Figure 6c). Mice treated with LT showed a slower tumor growth, whilst mice treated with LDT showed a marked growth delay (Figure 6d). To corroborate the delay in tumor growth, the time average that the tumors took to reach the 25%, 50%, and 75% of the final volume at the moment of the sacrifice in every group was analyzed (Figure 6e). Although no differences were observed in the time average at which the control and LT groups reached the 25% and 50% of the tumor growth, mice treated with LT presented a significant delay in the time they took to reach 75% of the tumor growth when compared to the control group. On the other hand, tumors of mice treated with LDT took significantly longer to reach 25%, 50%, and 75% of the tumor volume when compared to both the control and LT groups.

## 3. Discussion

TRAIL-based therapies in cancer exhibited very limited therapeutic activity in clinical trials [6,7,8,9], therefore, novel TRAIL formulations with improved bioactivity that can overcome TRAIL resistance would be of utility as cancer treatment [21,22,23,24,25,26]. In this regard, we have demonstrated in the present study that LDT exhibited a greatly improved cytotoxicity against a broad panel of cancer cells from distinct origins, overcoming resistance to sTRAIL or to LT, while LD alone did not show a remarkable cytotoxic effect in the case of epithelial cells, in which cytotoxicity was dependent on the presence of TRAIL. In the case of sarcoma cells, LD contributed to the cytotoxic effect, but the presence of TRAIL enhanced cytotoxicity even more. Finally, in hematological cancer cell lines, with the exception of MOLT4 cells, in which the greater cytotoxicity observed with LDT seemed to be attributable rather to DOX, cytotoxicity was due to the combination of TRAIL and DOX. In any case, the use of TRAIL and DOX in a unique formulation such as LDT exhibited an enhanced cytotoxic effect when compared to LT or LD alone on all cell lines tested, overcoming the resistance to sTRAIL. In this line, soluble DOX has been long described to synergize with sTRAIL when administered in combination [53,54,55,63,64]. However, this synergy happened at minimum incubation times of at least 12 hours, while the effect observed with LDT happened in a much shorter time frame. Moreover, a synergistic effect of LT with soluble DOX at 64.56 μM cannot be achieved. However, the effective amount of DOX that would actually get inside the cells when liposomal DOX was used might be much higher. In other words, liposomal DOX might be more effective than soluble DOX entering the cells, so the effective intracellular concentration of DOX could be much higher when liposomal DOX were used. Thereby, to observe a theoretical synergistic effect of soluble DOX with LT similar to that occurring in LDT, the concentration of soluble DOX should be much higher. However, when LT were used in combination with soluble DOX with increasing concentrations of soluble DOX up to a maximum of 1 mM, no cytotoxic effect was observed, ruling out any synergistic effect of LT with soluble DOX. Moreover, although previous studies have shown that DOX synergizes with soluble TRAIL to kill cancer cells, inhibition of cell death by the blocking antibody RIK clearly reflected a pivotal role for TRAIL in LDT-mediated cytotoxicity and was in agreement with the fact that LD alone had a rather low cytotoxicity. Besides, the data obtained clearly indicates that the cell death induced by LDT-is a classic caspase-dependent apoptotic process, relying on TRAIL-mediated caspase-8 activation.

Then, the involvement of the endocytic pathway in LD- and/or LDT-induced cell death was analyzed, trying to ascertain the mechanism underlying the enhanced cytotoxic effect exhibited by LDT. In this regard, dynamin-dependent endocytosis has been described to be involved in the cellular uptake of the liposomal DOX formulation Doxil^®^ [60]. Therefore, we blocked dynamin-dependent endocytosis by using the specific dynamin inhibitor Dynasore (DYN) [61]. Indeed, DYN was able to block endocytosis of both liposomal formulations, and especially cell death induced by LDT. Moreover, both LD and LDT appeared to be internalized in a similar manner and stored at lysosomes, although with different kinetics. The differences between both liposomal formulations would arise at that point, where the apoptotic signal triggered by liposomal TRAIL would somehow act in synergy with DOX located inside the lysosomes. According to this scenario, a situation in which LT were administered in combination with LD should render a similar outcome to LDT. In fact, although a similar effect could be obtained by treating the cells with LD and LT in parallel, LDT presents the advantage of assuring the simultaneous action of both agents (DOX and TRAIL), maximizing the pro-apoptotic efficiency. Taking the results altogether, the cytotoxicity induced by LDT was caspase-8- and TRAIL-dependent and involved an endocytosis mechanism dependent of dynamin. The mechanism proposed to explain the cytotoxic effect of LDT is summarized in Figure 7.

First, a dynamin-dependent endocytosis of the liposome-encapsulated DOX occurred. In fact, microscopy and flow cytometry results showed that LD was captured by the cells independently of the presence of TRAIL on the liposomal surface. In line with this, the commercial liposomal DOX formulations used in cancer treatment have been proven to be internalized by dynamin-dependent mechanisms [60]. Hence, both LD and LDT would end up sequestered in lysosomes after being endocytosed. It is precisely at this point when the caspase-8 activated by the liposome-displayed TRAIL would come into action by triggering the lysosomal rupture, facilitating the release of free DOX to the cytosol, and finally, its translocation to the nucleus, unleashing a potent apoptotic process. In fact, lysosomal permeabilization by active caspase-8 has already been described in the literature [65,66,67]. Interestingly, this model is only valid for liposome-displayed TRAIL, since soluble TRAIL did not show any synergistic effect when administered in parallel with LD (data not shown). This would highlight the need for a robust caspase-8 activation, which could only be provided by LT or LDT.

Novel liposomal formulation anchoring TRAIL on the surface and encapsulating DOX inside lumen, that is LDT, exhibited an extraordinary cytotoxic effect against a wide range of tumor types; however, it was also cytotoxic on freshly isolated PBMC and on activated T lymphocytes. Therefore, LDT with a concentration of DOX 10 times less than those previously described but with the same concentration of TRAIL, named LDT 1/10, was also generated in order to diminish LDT toxicity on normal cells. In fact, LDT 1/10 was not toxic on neither PBMC nor activated T lymphocytes. These results agree with previous data indicating that neither sTRAIL nor LT induced significant cell death on T-cells [37,46,68], and demonstrate that LDT with a lower concentration of DOX did not cause cytotoxic effects on normal cells. However, the reduction of DOX concentration into LDT could hamper their cytotoxicity against cancer cells. As expected, LDT 1/10 exhibited slightly less cytotoxicity than LDT, though the reduction was only minute, and all cells were still largely sensitive to the 1/10 version of LDT. All these data, taken together, indicate that optimization of the concentration of DOX encapsulated inside the liposomal lumen allows it to retain its cytotoxicity against tumor cells but suppress toxicity for normal cells.

Finally, we wanted to assess the in vivo anti-tumor activity of LDT. First, and before starting any in vivo approximation, was a possible systemic toxicity exerted by LDT 1/10. In fact, his-tagged sTRAIL was shown to have a cytotoxic effect on primary human hepatocytes [69,70]. Therefore, a pilot study was designed seeking to analyze LDT 1/10 in vivo safety. Serum obtained by blood samples from mice treated with LDT 1/10 was analyzed to determine the levels of aspartate aminotransferase (AST) and alanine aminotransferase (ALT), clinical markers for liver health, by standard protocols, as previously described [57]. At the same time, liver, kidney, spleen, and ovaries were also surgically extracted to perform a histological analysis of any possible tissular damage induced by LDT 1/10. The analysis of serum levels of the transaminases alanine aminotransferase (ALT) and aspartate aminotransferase (AST) did not show relevant changes when comparing LDT 1/10 treated mice with control mice. In addition, the normal tissue architecture was conserved in all tissues analyzed, including hepatic tissue. No signs of inflammatory infiltrate, fibrosis, or necrosis could be detected in any tissue in mice treated with LDT 1/10. Therefore, it could be concluded that LDT 1/10 did not induce any systemic toxicity on any tissue analyzed, validating their use in vivo. Once any systemic toxicity of LDT 1/10 was discarded, their anti-tumor activity was analyzed in an in vivo model using HT1080-xenograft tumors in nude mice. LDT 1/10 was able to slow down the tumor growth in a significant manner when compared to untreated animals and most importantly, when compared to LT. These results confirmed that LDT not only showed high in vitro bioactivity but also a strong in vivo anti-tumor activity.

## 4. Materials and Methods

### 4.1. Generation of Different Versions of Lipid Nanoparticles Decorated with TRAIL

Large unilamellar vesicles (LUVs) with soluble TRAIL (sTRAIL) tethered on their surface (hereafter LT) were generated as previously described [37,57]. Briefly, LUV-type lipid nanoparticles were generated with the following lipid composition: phosphatidylcholine, sphingomyelin, cholesterol, 1,2-distearoyl-sn-glycero-3-phosphoethanolamine)-N-(methoxy(polyethylene glycol)-2000) (ammonium salt) (18:0 PEG2000-PE), and 1,2-dioleoyl-sn-glycero-3-((*N*-(5-amino-1-carboxypentyl)-iminodiacetic acid)succinyl) (nickel salt) (DOGS-NTA-Ni) (Avanti Polar Lipids) with a weight ratio of 50:30:10:5:5. After that, sTRAIL (rTRAIL-His_6_, corresponding to amino acids 95–281, kindly provided by Dr. M. MacFarlane [71]) was attached to their surface by incubation at 37 °C for 30 minutes.

Large unilamellar vesicles (LUVs) with DOX encapsulated inside their lumen and decorated with TRAIL on their surface (hereafter LDT, Figure 1a) were synthetized as outlined in the protocol shown in Figure 1b. The method used for generation of LDT was the pH gradient method [72,73]. First, LUVs were prepared by using 300 mM (NH_4_)_2_SO_4_ (ammonium sulphate, AS) at pH = 4 to resuspend the dry lipid film for generating LUV-AS. Once extruded, LUV-AS were filtrated through a PD-10 desalting column (GE Healthcare) to create a pH and ion gradient between the external medium and the liposomal lumen by exchanging the external buffer to PBS. After that, LUV-AS were incubated with DOX at a molar drug-to-lipid ratio of 1/3.8 at 60 °C for 30 minutes, shaking at 800 rpm for obtaining LUV with DOX encapsulated inside (LUVDOX, hereafter LD), and then they were incubated with sTRAIL to obtain LDT. The final concentrations of total lipid, TRAIL, and DOX were 2.5 mM, 12 μg/mL and 775 μM, respectively. LDT with a concentration of DOX 10 times less than those previously described (LDT 1/10) was also generated. To assess the encapsulation efficiency of DOX inside the liposomal lumen, the absorbance of DOX at 480 nM was measured and a calibration curve using free DOX (concentration range from 0 to 40 μM, Appendix A) was used to finally quantify the amount of encapsulated DOX. Samples were taken from the reaction mixture at the indicated times and were subjected to size exclusion filtration through PD-10 desalting columns to capture the free doxorubicin. The flow through fractions containing the liposomes were collected and lysed in PBS containing 1% Triton X-100 (Sigma, St. Louis, MS, U.S.) to disrupt the lipid bilayer and release the entrapped DOX. Absorbance at 480 nm of each fraction was measured. As a negative control, the encapsulation process was also carried out with classic LUV (containing PBS instead of AS) in parallel.

Once the feasibility and efficiency of DOX encapsulation inside LUV-AS (LUVDOX) was proven, the next step was to attach sTRAIL on their surface. For that, sTRAIL was incubated with LD at the same protein/lipid ratio as in the classic LT [37] to obtain the LDT formulation. Since sTRAIL was coupled to LD incubating at 15 °C to diminish a possible DOX leakage, instead of 37 °C, as in the case of classic LT formulation, the efficiency of the TRAIL attachment process was checked to discard a possible low-binding efficiency. For that, LDT were ultra-centrifuged at 60,000 rpm for 5 hours, and the presence of uncoupled 25 kD as TRAIL in the supernatant was assessed by Western blot (Appendix A). The coupling efficiency was very high and similar to that of classic LT [57], validating the modification of the protocol. Therefore, the final concentration of DOX and TRAIL in the stock of LDT was 775 μM and 12 μg/mL, respectively.

### 4.2. Cell Culture

A549 (derived from non-small cell lung carcinoma), HT-29 (derived from colon adenocarcinoma), RAJI (derived from Burkitt’s lymphoma), U937 (derived from histiocytic lymphoma), MOLT4 (derive from T-cell acute lymphoblastic leukemia), A673 (derived from Ewing’s sarcoma), HT-1080 (derived from fibrosarcoma), and RD cells (derived from rhabdomyosarcoma) cell lines were obtained from ATCC. Cell lines were routinely cultured at 37 °C with 5% CO_2_ in DMEM medium supplemented with 10% fetal calf serum (FCS), 2 mM L-glutamine, and penicillin/streptomycin (hereafter, complete medium).

Peripheral blood mononuclear cells (PBMC) were obtained from whole peripheral blood samples from healthy donors by density gradient centrifugation using Ficoll-Paque^®^ (GE Healthcare). T-cell blasts were generated by stimulating PBMC (2 × 10^6^ cells/mL) with 5 µg/mL PHA (phytohemagglutinin) for 1 day. Then, PHA was removed, and cells were resuspended in complete medium supplemented with 30 UI/mL IL-2 and cultured for 6 days changing medium every 48 h.

### 4.3. Cytotoxicity Assays and Apoptosis Quantification

Tumor cell lines (3 × 10^4^ cells) were treated with LT and LDT for 24 hours at the following concentrations: 1, 4, 12, 37, 11, 333, and 1000 ng/mL of TRAIL, which corresponded to the following concentrations of DOX: 0.09, 0.27, 0.8, 2.39, 7.17, 21.52, and 64.56 µM, respectively. Dose-response experiments were also performed using LDT with the same concentration of TRAIL, but with a concentration of DOX 10 times less than those previously indicated (LDT 1/10). LUV-type liposomes without TRAIL anchored to their surface but loaded with DOX (LD and LD 1/10) with the previously described doses were also used in cytotoxicity assays. Time-course experiments were also carried out incubating the epithelial-derived cancer cell lines A549, SKBR3, and HT-29 and the sarcoma cell lines A673, HT-1080, and RD with 1000 ng/mL of LT, LD, and LDT for 1, 2, 4, and 6 hours. Times of incubation at 8 and 20 hours were also included for sarcoma cells. Experiments of apoptosis inhibition assays were carried out by pre-incubating cells with the blocking anti-human TRAIL mAb (500 ng/mL, clone RIK2, BD Biosciences), with the pan-caspase inhibitor z-VAD-fmk (30 µM, Bachem, Bubendorf BL, Switzerland) or with the specific caspase-o inhibitor IETD-fmk (30 µM, Bachem) for 1 hour prior to the addition of LT or LDT. Finally, apoptosis induced by different liposome formulations was analyzed by annexin-V staining using a FACSCalibur flow cytometer and CellQuest software (BD Biosciences).

### 4.4. Scanning Electron Microscopy

Samples of LDT for analysis by SEM were prepared as previously described [57]. Briefly, after adding glutaraldehyde to bare LDT suspensions up to a final concentration of 1%, a drop was placed onto a glass microscope slide previously treated with 3-aminopropyltriethoxysilane (Sigma). Then, the sample was washed with PBS, fixed with 1% OsO4, and washed again with PBS. Thereafter, the sample was dehydrated in a graded series of ethanol (25%–100%), mounted on specimen, and coated with gold. Finally, the sample was analyzed using a Hitachi S-3400 N scanning electron microscope (Hitachi, Tokyo, Japan).

### 4.5. Western Blot Analysis

Western blot was performed for protein expression analysis, as previously described [47,48,49]. The antibodies used to analyze the expression of the main proteins involved in the apoptotic pathway were: anti-caspase-8 (BD Biosciences), anti-caspase-3 (Cell Signaling), anti-Bid (BD Biosciences), anti-PARP-1 (BD Biosciences), and anti-caspase-9 (MBL). The expression of β-actin was analyzed as loading control, using a specific antibody (Sigma). Western blot was also carried out to assess the binding of TRAIL to LD using the anti-TRAIL antibody RIK2 (eBiosciencies).

For Western blot analysis, cells (5 × 10^6^) were lysed at 4 °C with 100 µL of a buffer containing 1% Triton X-100 and protease and phosphatase inhibitors, as previously described [46,47,48,49]. Then, protein from lysed cells were separated by 12% SDS-PAGE, transferred to PVDF membranes, and blocked with TBS-T buffer (10 mM Tris/HCl, pH 8.0, 0.12 M NaCl, 0.1% Tween-20, 0.05% sodium azide) containing 5% skimmed milk. PVDF membranes were incubated with the above-mentioned antibodies in TBS-T containing 2% skimmed milk. After that, PVDF membranes were washed with TBS-T, incubated with 0.2 µg/mL of peroxidase-labeled or phosphatase alkaline-labeled secondary antibody (Sigma), and revealed with the corresponding substrate Pierce ECL Western Blotting Substrate (Thermo Scientific, Waltham, MA, U.S.) or CDP-Star (Merck, Kenilworth, New Jersey, U.S.), respectively.

### 4.6. Fluorescence Microscopy Analysis

Analysis of the intracellular location of liposome formulation were performed by fluorescence microscopy. For that, A549 cells were pre-treated for 1 h with Lysotracker™ green (ThermoFisher). Then, cells were treated with LD or LDT at their maximum working concentrations (1000 ng/mL TRAIL; 64.56 μM DOX) for 1 hour. At the final time, medium was carefully replaced, and cells were cooled for 10 min at 4 °C. Finally, green fluorescence corresponding to lysosomal compartment and red fluorescence corresponding to DOX were photographed using a fluorescence microscopy (E600/E400, Nikon, Tokyo, Japan) equipped with a digital photograph system (DXM 1200F, Nikon).

### 4.7. Confocal Fluorescence Microscopy Analysis

For analysis of LD and LDT by confocal microscopy, A549 cells were seeded in 8-well chamber slides (Labtek^®^, 20 × 10^4^ cells per condition). The following day, cells were treated for the indicated times with LD or LDT. At the end of the treatment, slides were quickly washed with cold PBS and fixed in PFA 4% for 30 minutes at room temperature in the dark. Next, cells were blocked with PBS 5% FCS, and then they were stained overnight with an anti-LAMP1 antibody (Cell Signaling). The following day, cells were washed and incubated with an Alexa 488 secondary antibody. Finally, cells were washed, and mounted onto drops of Fluoromount-G (Southern Biotech) containing DAPI. Fluorescence images were taken using a confocal microscopy (Olympus FV10i). Data was analyzed using a FV10-ASW Viewer, v3.1 (Olympus).

### 4.8. Study of the Internalization Mechanism

To analyze the endocytic pathways involved in LT-induced signaling, Dynasore (DYN, Sigma), a small molecule GTPase inhibitor that targets dynamin-1, -2, and Drp1, was used [61]. The cell lines A549, HT-29, and SKBR3 were incubated in the presence or absence of DYN (80 μM) for 1 hour. After that, cells were treated with LD or LDT at 1000 ng/mL for 3 hours. Experiments pre-incubating cells with the caspase-8 inhibitor z-IETD-fmk (30 µM) for 1 hour prior to addition of liposome formulations were also performed. Finally, apoptosis was quantified by annexin-V staining. Doxorubicin uptake was measured quantifying the mean fluorescence intensity (MFI) at the FL3 channel (red fluorescence). Both annexin-V staining and DOX MFI were analyzed using a FACSCalibur flow cytometer and CellQuest software (BD Biosciences).

### 4.9. In Vivo Study of Toxicity

Immune-deficient athymic mice, Hsd:Athymic Nude-Foxn1^nu^, six-week-old females (Janvier), were used to evaluate *in vivo* toxicity. All experiments in mice were carried out according to the European recommendations on animal ethics and were approved by the University of Zaragoza Animal Experimentation Ethical Committee (PI 18/16).

In order to assess the toxicity of LDT, five animals were systemically injected with 36 µg of LDT 1/10 4 times with an interval of 2 days between each injection. Control mice (2 animals) were injected with PBS. After 2 weeks, mice were sacrificed and blood was collected by heart puncture, and serum was obtained from them by centrifugation and stored at −20 °C until analysis. Aspartate aminotransferase (AST) and alanine aminotransferase (ALT) serum levels were determined as previously described [48,57]. At the same time, mice were sacrificed, and liver, kidney, spleen, and ovaries were surgically dissected and fixed in 10% buffered formalin. After fixation for three days, tissues were embedded in paraffin and sectioned in 5 µm sections. The slides were then deparafined in xylene, dehydrated in a gradient of alcohols, and stained using hematoxylin and eosin (H and E).

### 4.10. In Vivo Anti-Tumor Activity

Same immune-deficient athymic mice as above mentioned (Hsd:Athymic Nude-Foxn1^nu^, six-week old females), were used to evaluate *in vivo* antitumor activity. As previously mentioned, all experiments in mice were performed following the European recommendations on animal ethics and were approved by the University of Zaragoza Animal Experimentation Ethical Committee (PI 18/16). Mice were kept under specific standard pathogen-free conditions throughout the study.

Freshly harvested H-1080 cells (5 × 10^6^ cells/mouse) were inoculated subcutaneously into mice. Once tumors reached a volume of 100 mm^3^, mice were randomly divided in 3 groups: Control (4 mice), LT (6 mice), and LDT (6 mice). 36 μg of TRAIL as LT or LDT 1/10, in a total volume of 200 μL, were injected intra-peritoneally daily during 4 consecutive days. The Control group were injected with sterile PBS. Tumors were left to grow until they reached 1000 mm^3^ or when clinical signs required euthanasia. During all of the experiments, tumor volumes were monitored daily, and tumor volume was calculated using the following formula: tumor volume = L × W × H (where L is length, W is width, and H is the height dimensions of the tumor, respectively).

### 4.11. Statistical Analysis

Computer-based statistical analysis was carried out using the Prism^®^ software program (GraphPad Software, San Diego, CA, U.S.). All values are expressed as mean ± standard deviation (SD) of at least 3 experiments. Statistical significance was analyzed by using Student’s *t*-test. A *p <* 0.05 was considered statistically significant.

## 5. Conclusions

Doxorubicin (DOX) has been widely described to synergize with sTRAIL when administered in combination. In this line, nanoparticles combining both molecules have already been developed [38,74,75,76,77]. Consequently, the generation of a unique formulation that includes both anti-cancer drugs, such as LUVDOX-TRAIL, could be of great interest to treat different types of cancer. Our liposomal formulation shows a cytotoxic potential and a synergy never observed in any TRAIL and DOX combination before. In summary, LDT exerted an enhanced anti-tumor effect not only in vitro but also in vivo without overt adverse effects and could potentially be clinically relevant to improve anti-tumor therapy against TRAIL-resistant cancers.

## 6. Patents

Alberto Anel and Luis Martinez-Lostao have filed a patent application (W02011020933) for the use of liposome-bound Apo2L/TRAIL.

## Figures and Tables

**Figure 1 cancers-11-01948-f001:**
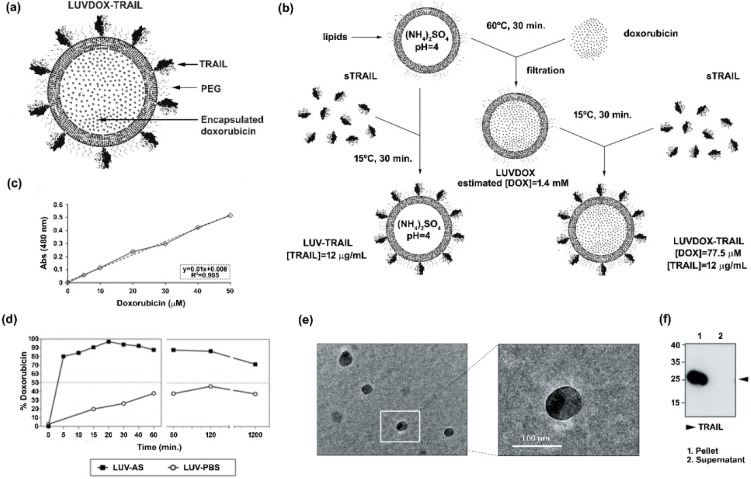
(**a**) Schematic representation of large unilamellar vesicle doxorubicin-TNF-related apoptosis-inducing ligand (LUVDOX-TRAIL (LDT)). (**b**) Schematic illustration of the generation of LDT. Generation of LDT was performed as describe in the Experimental Section. (**c**) Standard curve of doxorubicin concentration. Absorbance at 480 nm of the indicated concentrations of doxorubicin (DOX) was measured. The DOX standard curve was used to interpolate the results obtained in (d). (**d**) Encapsulation efficiency and release profile of doxorubicin in LUV-AS. DOX was incubated with LUV-AS at a molar ratio of 1:3.8 (drug:lipid) for 30 min at 60 °C. At the indicated times, samples were taken and free DOX was separated from the liposomal fraction. Encapsulated DOX was quantified by measurement of absorbance at 480 nm. Efficiency is represented as percentage of total doxorubicin present in the liposomal fraction in every time point. (**e**) Scanning electron microscopy of LDT was carried out as described in the Experimental Section. Original magnification was 45,000×. (**f**) Coupling efficiency of soluble TRAIL in the LDT formulation. Once LUVDOX-TRAIL were generated, they were ultra-centrifuged at 60,000 rpm for 5 h, collecting the supernatant and resuspending the pellet. Aliquots from the pellet and the supernatant fractions were separated by SDS-PAGE and the presence of TRAIL in both fractions was assessed by Western blot.

**Figure 2 cancers-11-01948-f002:**
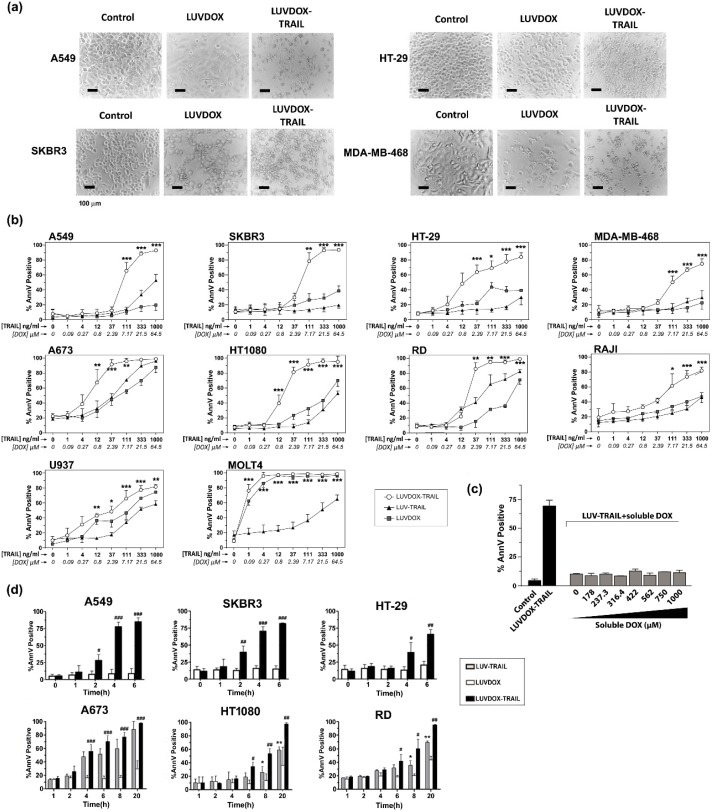
(**a**) Cytotoxicity of LUVDOX-TRAIL (LDT) on a panel of tumor cell lines from distinct origin. A549, HT-29, SKBR3, and MDA-MB-468 cells were treated with LUV-TRAIL (LT), LUVDOX (LD), or LDT (TRAIL at 1 µg/mL) for 24 h. Bright field microscopy images of the Control, and cells treated with LD and LDT, were taken. Original magnification was 200×. (**b**) A panel of tumor cell lines (A549, HT-29, SKBR3, MDA-MB-468, A673, HT-1080, Raji, RD, U937, and MOLT4) were treated with increasing concentrations of LT, LD, or LDT for 24 h. The following day, cell death was measured by annexin-V staining. Graphs show the mean ± standard deviation (SD) of at least three independent experiments. * *p* < 0.05, ** *p* < 0.01, *** *p* < 0.001. (**c**) Combined treatment of LT with increasing concentrations of soluble DOX on A549 cells. A549 cells were treated with LT (1000 ng/mL) in combination with increasing concentrations of soluble DOX for 3 h. Besides, LDT was also used as a reference. Results are the mean ± SD of three independent experiments. (**d**) Time-course cytotoxicity of LDT on the tumor cell cells: A549, SKBR3, HT-29, A673, HT-1080, and RD cells. Cells were treated with LD or LDT at their maximum concentrations (1 μg/mL TRAIL; 64.56 μM DOX) for the indicated times. Apoptotic cells were measured by annexin-V staining. Graphs show the mean ± SD of four different experiments. * *p* < 0.05, ** *p* < 0.01, LT versus LDT ^#^
*p* < 0.05, ^##^
*p* < 0.01, ^###^
*p* < 0.001 LD versus LDT.

**Figure 3 cancers-11-01948-f003:**
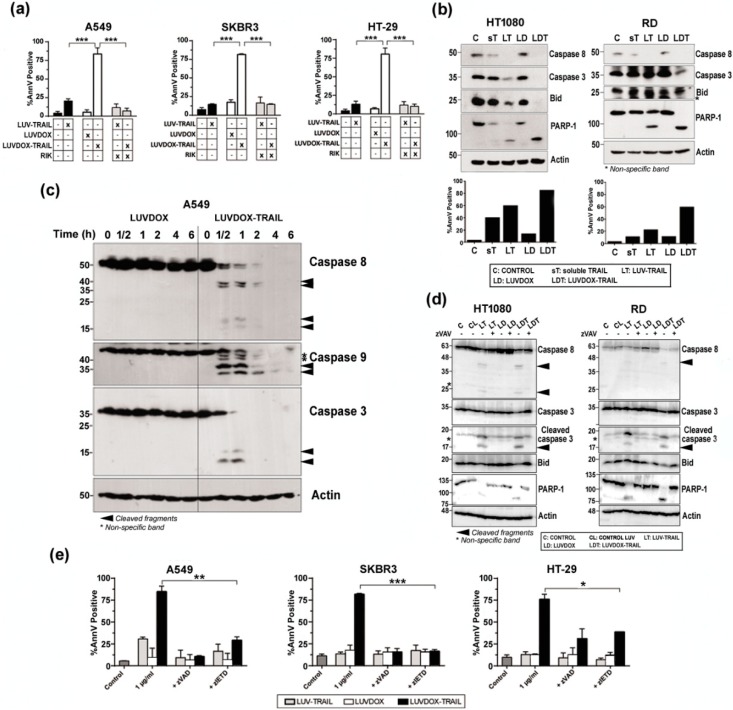
(**a**) Roles of DOX and TRAIL in LUVDOX-TRAIL (LDT) cytotoxicity. A549, SKBR3, and HT-29 cells were treated with the indicated combinations of LUV-TRAIL (LT), LUVDOX (LD), and LDT (TRAIL at 1000 ng/mL). When indicated, cells were pre-treated with the TRAIL-blocking antibody RIK. Cell death was assessed by annexin-V staining after 6 h of treatment. Graphs show the mean ± SD of at least four experiments. *** *p* < 0.001. (**b**) Analysis of caspase activation after treatment with different TRAIL versions. HT-1080 and RD cells were untreated (Control, designed as C), or treated with soluble TRAIL (ST), LT, LD, and LDT at 1000 ng/mL. After 24 h, cells were lysed, and lysates were subjected to SDS-PAGE and to Western blot analysis. Levels of caspase-8, caspase-3, Bid, and PARP-1 were analyzed using specific antibodies. The level of actin levels was used as a control for equal protein loading. Cell death was quantified in parallel by flow cytometry after annexin-V staining (bottom panels). (**c**) Analysis of time-course caspase activation with LD or LDT. A549 cells were treated with LD or LDT (1 μg/mL TRAIL; 64.56 μM DOX) at the indicated times. Finally, cells were lysed, and lysates were subjected to SDS-PAGE and to Western blot analysis. (**d**) Analysis of caspase activation after treatment with different TRAIL versions. HT-1080 and RD cells were untreated (Control, designed as C), or treated with LUV alone (CL), LT, LD, and LDT at 1000 ng/mL. When indicated, cells were pre-treated with the pan-caspase inhibitor z-VAD-fmk (30 μM). After 24 h, cells were lysed, and lysates were subjected to SDS-PAGE and to Western blot analysis. Levels of caspase-8, caspase-3, Bid, and PARP-1 were analyzed using specific antibodies. Actin levels was used as a control for equal protein loading. (**e**) Role of caspases in LDT cytotoxicity. A549, SKBR3, and HT-29 cells were treated with LT, LD, or LDT (TRAIL at 1000 ng/mL). When indicated, cells were pre-treated either with the pan-caspase inhibitor z-VAD-fmk (30 μM) or with the specific caspase-8 inhibitor z-IETD-fmk (30 μM) for 1 hour. Cell death was assessed by annexin-V staining after 6 h of treatment. Graphs show the mean ± SD of at least four experiments. * *p* < 0.05, ** *p* < 0.01, *** *p* < 0.001.

**Figure 4 cancers-11-01948-f004:**
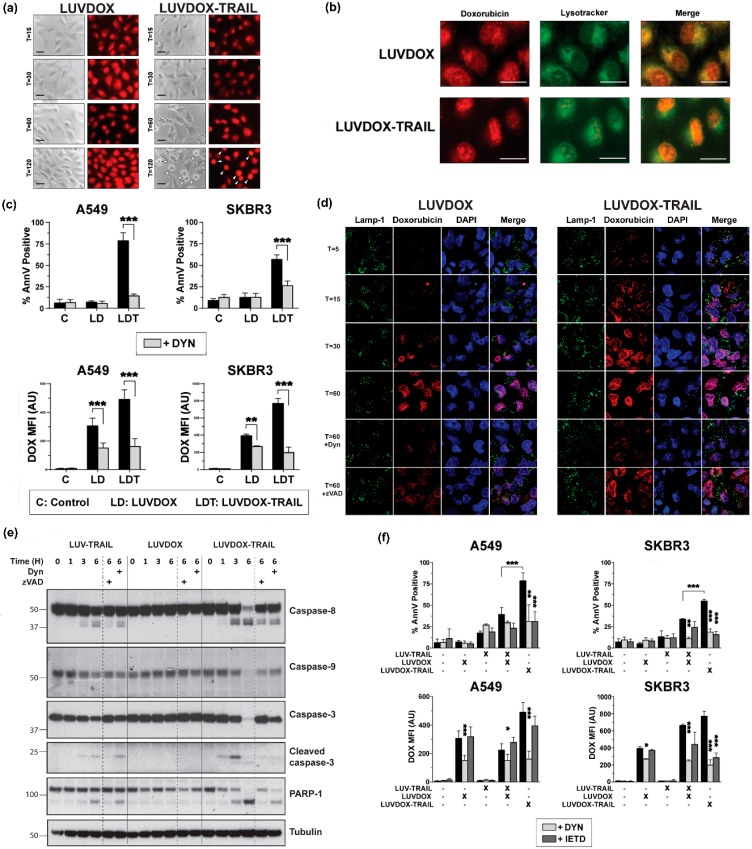
(**a**) Fluorescence microscopy of A549 upon treatment with LD or LDT. A549 cells were treated with LD or LDL at their maximum working concentrations (1 μg/mL TRAIL; 64.56 μM DOX) at the indicated times. At the final time, medium was carefully replaced, and cells were cooled for 10 minutes at 4 °C. Finally, bright field and red fluorescence pictures were taken by fluorescence microscopy. (**b**) Subcellular location of LD and LDT. A549 cells were pre-treated for 1 h with Lysotracker™. Then, cells were treated with LD or LUVDOX-TRAIL (TRAIL, 1000 ng/mL TRAIL) for 1 h. At the final time, medium was replaced, and cells were cooled for 10 min at 4 °C. Finally, green and red fluorescence pictures were taken by fluorescence microscopy. Original magnification was 400×. (**c**) Inhibitory effect of dynasore on cell death and DOX fluorescence of upon treatment with LD or LDT. A549 and SKBR3 cells were treated with LD or LDT for 3 h. In some cases, cells were pre-treated with dynasore (DYN, 80 μM) for 1 h. Cell death was measured by annexin-V staining and DOX uptake was quantified by red-fluorescence measurement, respectively. Graphs show the mean ± SD of four independent experiments. * *p* < 0.05, ** *p* < 0.01, *** *p* < 0.001. (**d**) Confocal microscopy of A549 upon treatment with LD or LDT. A549 cells were treated with LD or LDL at their maximum working concentrations (1 μg/mL TRAIL; 64.56 μM DOX) at the indicated times. In some cases, cells were pre-treated with dynasore (DYN, 80 μM) or z-VAD-fmk (zVAD, 30 μM) for 1 hour. Cells were stained with antibody anti-LAMP-1 and the probe DAPI. At the final time, medium was carefully replaced, and cells were cooled for 10 minutes at 4 °C. Finally, fluorescence pictures were taken by confocal microscopy. (**e**) Analysis of the time-course activation of caspases after treatment with LT, LD, and LDT. A549 cells were treated with LT, LD, and LDT at 1000 ng/mL. When indicated, cells were pre-treated either with DYN (80 μM) or z-VAD-fmk (30 μM). After, cells were lysed, and lysates were subjected to SDS-PAGE and to Western blot analysis. Levels of caspase-8, caspase-9, caspase-3, and PARP-1 were analyzed using specific antibodies. Level of actin was used as a control for equal protein loading. (**f**) Effect of combined treatment of LD and LUV-TRAIL on cell-death and DOX-fluorescence. Tumor cells were treated with LD and LUV-TRAIL alone, or with the combination of both LD and LUV-TRAIL for 3 h. Besides, LDT was used in parallel as a reference. In some cases, cells were pre-treated with dynasore (DYN, 80 μM) or z-IETD-fmk (zIETD, 30 μM) for 1 hour. (e) Cell death was measured by annexin-V staining and (f) DOX uptake was quantified by red-fluorescence measurement, respectively. Graphs show the mean ± SD of five independent experiments. * *p* < 0.05, ** *p* < 0.01, *** *p* < 0.001.

**Figure 5 cancers-11-01948-f005:**
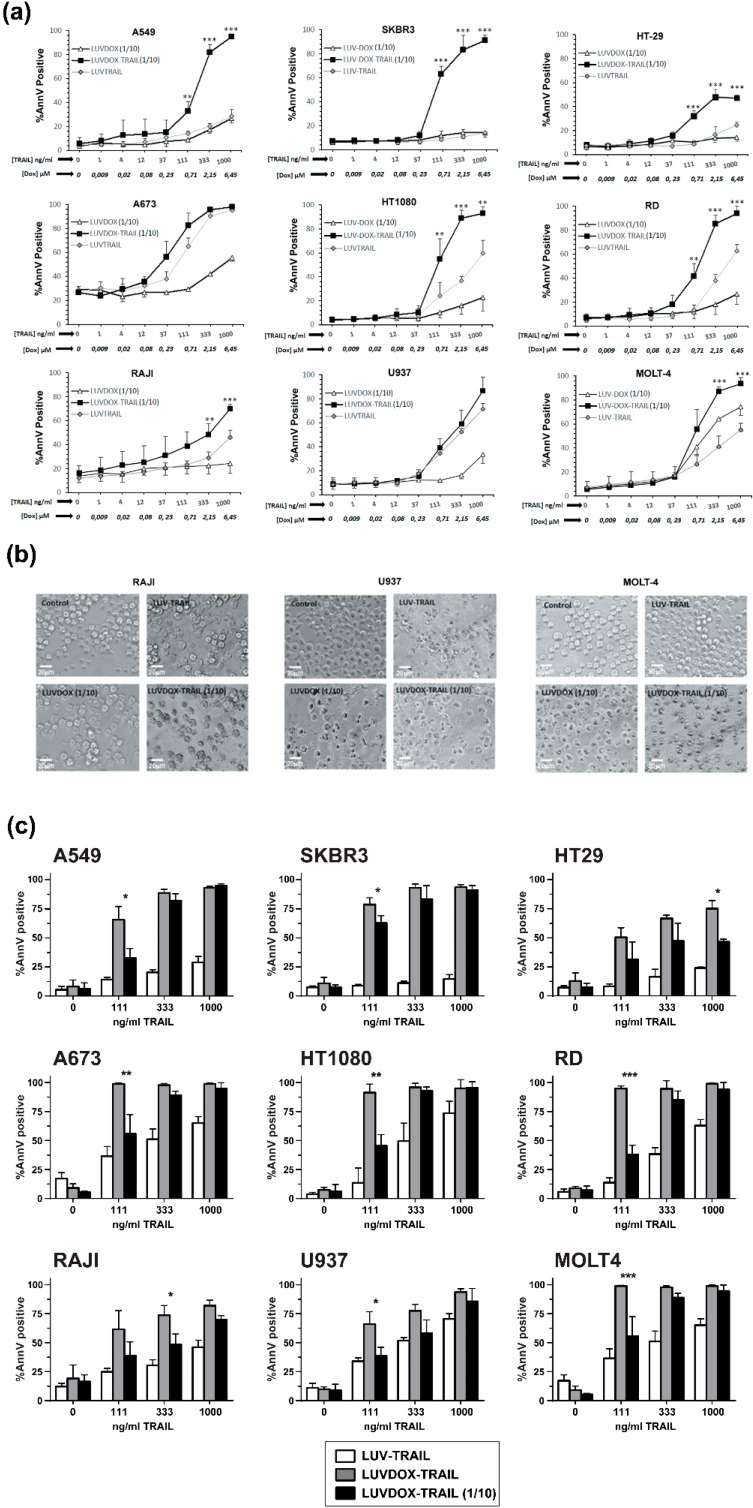
(**a**) Cytotoxicity of LUVDOX-TRAIL 1/10 (LDT 1/10) on a panel of tumor cell lines from distinct origin. A panel of tumor cell lines (A549, HT-29, SKBR3, A673, HT-1080, Raji, RD, U937, and MOLT4) were treated with increasing concentrations of LUV-TRAIL (LT), LUVDOX 1/10 (LD 1/10), or LDT 1/10 with entrapped DOX concentration of DOX: 6.45 Μm (DOX 1/10). After 24 h, cell death was measured by annexin-V staining. Graphs show the mean ± SD of at least three independent experiments. * *p* < 0.05, ** *p* < 0.01, *** *p* < 0.001. (**b**) Cytotoxicity of LDT 1/10 on a panel of tumor cell lines from distinct origin. Raji, U937, and MOLT-4 cells were treated with LT, LD 1/10, or LDT 1/10 for 24 h. Bright field microscopy images of the Control, and cells treated with LD and LDT, were taken. Original magnification was 200×. (**c**) Comparison of LDT and LDT 1/10 on a panel of tumor cell lines from distinct origin. A panel of tumor cell lines (A549, HT-29, SKBR3, A673, HT-1080, Raji, RD, U937, and MOLT4) were treated with LT, LDT, and LDT 1/10 with different entrapped DOX concentrations (LDT: DOX at 64.56 μM or LDT 1/10: DOX at 6.45 μM). The final concentration of TRAIL was 1000 ng/mL in all cases. After 24 h, cell death was quantified by annexin-V staining. Graphs show the mean ± SD of at least three independent experiments. * *p* < 0.05, ** *p* < 0.01, *** *p* < 0.001.

**Figure 6 cancers-11-01948-f006:**
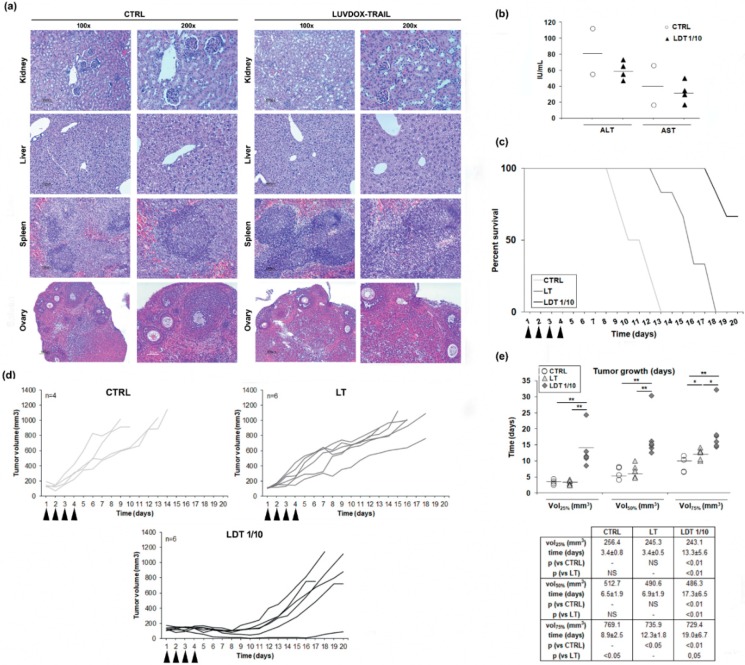
(**a**) Histological analysis of different tissues derived from LUV-DOX-TRAIL 1/10-treated mice. Mice were subjected to systemic LUVDOX-TRAIL 1/10 (LDT 1/10) treatment 4 times with an interval of 2 days. After that, organs were surgically dissected and fixed in 10% buffered formalin. After fixation for 3 days, tissues were embedded in paraffin and sectioned in 5 μm sections. The slides were then deparafined in xylene, dehydrated in a gradient of alcohols, and stained using hematoxylin and eosin (H and E). Original magnification was at 100× and 200×. (**b**) Determination of serum levels of AST and ALT in LDT 1/10-treated mice. Mice were subjected to systemic LUV-TRAIL treatment 4 times with an interval of 2 days. After that, mice were sacrificed, and blood samples were collected by heart puncture. Blood samples were then centrifuged to obtain serum, and AST and ALT levels were analyzed in the obtained serum samples by nephelometry. Graphics represent the results as a dot plot of the data collected from 4 mice in the LDT 1/10 group and 2 mice in the control group. (**c**) Mice bearing HT-1080 tumors received intra-peritoneal injections of PBS, LUV-TRAIL (LT, 36 μg of TRAIL/injection), or LUVDOX-TRAIL 1/10 (LDT 1/10, 36 μg of TRAIL/injection) for 4 days (arrow-pointed). Percent survival of mice from the control, LT, and LDT 1/10 groups was monitored for 20 days. Results are expressed as a Kaplan-Meier graphic. (**d**) In vivo anti-tumor activity of LUV-TRAIL and LUVDOX-TRAIL 1/10. Mice bearing HT-1080 tumors received intra-peritoneal injections of PBS, LUV-TRAIL (LT, 36 μg of TRAIL/injection), or LUVDOX-TRAIL 1/10 (LDT 1/10, 36 μg of TRAIL /injection) for 4 days (arrow-pointed). Tumor growth is depicted, calculated using the formula: tumor volume = L × W × H (where L is length, W is width, and H is the height dimensions of the tumor respectively). (**e**) Time-course analysis of the tumor growth in mice bearing HT-1080 tumors. Tumor volumes corresponding to 25%, 50%, and 75% respectively, of the final volume at the moment of the sacrifice in every group were calculated. Time (in days) at which tumors reached these volumes in every group was calculated and expressed as dot plot of animals in each group. * *p* < 0.05, ** *p* < 0.01.

**Figure 7 cancers-11-01948-f007:**
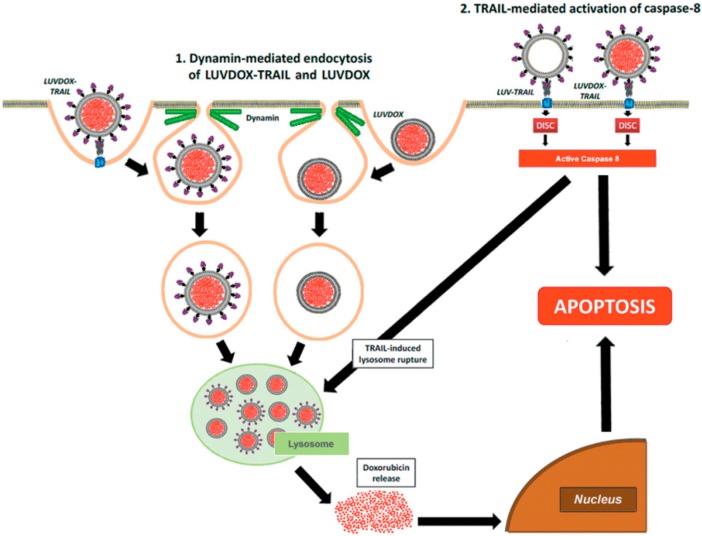
Schematic illustration of the mechanism proposed for cytotoxic effect of LUVDOX-TRAIL.

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
