# Peer review of "Double-Edged Lipid Nanoparticles Combining Liposome-Bound TRAIL and Encapsulated Doxorubicin Showing an Extraordinary Synergistic Pro-Apoptotic Potential"

_cancers, 2019, doi:10.3390/cancers11121948_

Round 1

Reviewer 1 Report

Indeed, Apo2L / TRAIL has shown significant anticancer effects in preclinical studies. At the same time, a sufficient effect was not found after the introduction of the protein to patients, possibly due to its insufficient activity. Cross-binding of antibodies significantly increased the cytotoxicity of soluble Apo2L / TRAIL against various cancer cell lines (Pradeep M. Nair, Heather Flores, Alvin Gogineni, Scot Marsters, David A. Lawrence, Robert F. Kelley, Hai Ngu, Meredith Sagolla, Laszlo Komuves , Richard Bourgon, Jeffrey Settleman, Avi Ashkenazi. PNAS. 2015. Vol. 112. No. 18. P. 5679-5684). The Apo2L / TRAIL protein presented on the surface of liposomes showed a significantly more pronounced anti-cancer effect in vivo.

In this regard, the relevance of the study is not in doubt. Nevertheless, I had a number of questions after a detailed review of the article:

Is the distribution of doxorubicin in the liposomes uniform? Spectral data give an average concentration over the entire fraction? Why, when considering the mechanism leading to apoptosis, were we limited to considering caspases 3, 8, and 9. Caspases 6, 7, and 10 were excluded for some reason at the preliminary stage? Why was doxorubicin chosen from the entire spectrum of chemotherapeutic drugs? Consider, for example, the proteasome inhibitor bortezomib or the cytostatic drug paclitaxel, since these drugs significantly enhance the induction of apoptosis by TRAIL in most resistant cell lines. Justification should be added to the introduction to the article. Why is only a 10-fold dose reduction considered? Perhaps this is not the best option. Explain your choice, please. The main problem when using anthracyclines (doxorubicin, epirubicin) is cardiotoxicity, characterized by irreversible damage to the heart muscle with a decrease in its contractile function, which can be manifested not only by a decrease in the ejection fraction of the left ventricle, but also by clinical signs of heart failure. Cardiotoxicity is cumulative and occurs when a certain total dose of the drug is reached. Liposomal doxorubicin does not reduce the effectiveness of the treatment; the toxicity spectrum remains similar, while cardiotoxicity phenomena are less pronounced. The section describing toxic effects does not mention the possible cardiotoxicity of this drug. Why? Many polyphenolic compounds, in most cases flavonoids, exhibit a synergistic effect with TRAIL, affecting various proteins involved in the regulation of apoptosis, survival or the rate of division of tumor cells. For example, quercetin can enhance the effect of TRAIL due to dephosphorylation of Akt and activation of caspases on human adenocarcinoma cells. No cytotoxicity was detected with respect to normal cells. Quercetin has been shown to be able to activate caspases-3, -8 and -9. This may be a promising area for your research. It should be added to the discussion of the results, as well as the options for using Apo2L / TRAIL in combination with other chemotherapeutic drugs. The therapeutic use of TRAIL is limited, since a significant portion of the tumor cells are insensitive to it. One of the reasons for this is the interaction of the cytokine with the receptor "traps" of DcRl and DcR2, which not only do not trigger the apoptotic signal, but also inhibit the signal of the death receptor. The authors obtained a unique DR5 selective mutant variant of TRAIL DR5-B, which binds to the DR5 receptor as effectively as the original cytokine, but practically does not bind to the DR4 and DcRl receptors, and its affinity for DcR2 is reduced by 400 times (ME Gasparian, et al. MUTATIONS ENHANCING SELECTIVITY OF ANTITUMOR CYTOKINE TRAIL TO DR5 RECEPTOR INCREASES ITS CYTOTOXICITY AGAINST TUMOR CELLS. Biochemistry. 2015; 80 (8): 1298-1311). In this work, we performed a comparative analysis of the cytotoxic activity of wild-type TRAIL and DR5-B on 12 different tumor cell lines and two types of normal cells. In nine out of 12 lines, DR5-B killed 1.5–5 times more tumor cells compared to TRAIL and did not show toxicity to normal cells. Chemotherapy drugs, such as doxorubicin, paclitaxel, and bortezomib, enhanced the effect of the TRAIL variants, but on most cell types the enhancement effect was more pronounced for DR5-B. The half-maximum effective concentrations (EC50) for DR5-B when combined with chemotherapy were 1.5-10 times lower compared to wild-type TRAIL. Thus, the DR5-B preparation can be considered as a promising drug, both for monotherapy and for combination w

The method described in the peer-reviewed article undoubtedly deserves attention, but the authors did not say anything about the limitations of the study and alternative options. Any chemotherapeutic drug will be toxic to healthy cells, including, despite dose adjustments. Consider using the mutant TRAIL DR5-B variant (or other option) in combination with your designs. Perhaps this will give the best therapeutic effect in reducing systemic toxicity.ith chemotherapy, in the treatment of TRAIL resistant tumors.

Author Response

Zaragoza, November 27, 2019

Cassidy Wang

Assistant Editor

Dear Editor,

Thank you very much for considering our manuscript for revision, giving us the opportunity to reply to the reviewers´ questions. Please, find enclosed a final copy of the revised version of our manuscript by De Miguel D. et al., entitled "Double-edged lipid nanoparticles combining liposome-bound TRAIL and encapsulated doxorubicin showing an extraordinary synergistic pro-apoptotic potential" (cancers-645176), and a point-by-point reply to reviewer’s suggestions. The authors would like to thank the work made by the reviewer on our paper, which have resulted in a net improvement of the initial version of the manuscript.

Nevertheless, before proceeding to discuss point by point the suggestions made by the reviewer 1, we would like to point out that given the short deadline we were given, we could only address experimentally some of the point raised by the reviewer.

Reviewer 1:

Indeed, Apo2L / TRAIL has shown significant anticancer effects in preclinical studies. At the same time, a sufficient effect was not found after the introduction of the protein to patients, possibly due to its insufficient activity. Cross-binding of antibodies significantly increased the cytotoxicity of soluble Apo2L/TRAIL against various cancer cell lines (Pradeep M. Nair, Heather Flores, Alvin Gogineni, Scot Marsters, David A. Lawrence, Robert F. Kelley, Hai Ngu, Meredith Sagolla, Laszlo Komuves , Richard Bourgon, Jeffrey Settleman, Avi Ashkenazi. PNAS. 2015. Vol. 112. No. 18. P. 5679-5684). The Apo2L/TRAIL protein presented on the surface of liposomes showed a significantly more pronounced anti-cancer effect in vivo.

In this regard, the relevance of the study is not in doubt. Nevertheless, I had a number of questions after a detailed review of the article:

Is the distribution of doxorubicin in the liposomes uniform? Spectral data give an average concentration over the entire fraction?

As described in Material and Methods, to assess the encapsulation efficiency of doxorubicin (DOX) inside the liposomal lumen, the absorbance of DOX at 480 nM was measured and calibration curve using free DOX (concentration range from 0 to 40 μM) was used to finally quantify the amount of encapsulated DOX. For that purpose, LUVDOX-TRAIL were taken from the reaction mixture, and were subjected to size exclusion filtration through PD-10 desalting columns to capture the free DOX. The flow through fractions containing the liposomes were collected and lysed in PBS containing 1% Triton X-100 to disrupt the lipid bilayer and release the entrapped DOX. Using this protocol, it can be assumed that the data obtained represents an average concentration over the entire fraction of liposomes with DOX encapsulated inside. In addition, the electron microscopy analyses revealed a similar distribution of the encapsulated DOX among the liposomal population. Of course, individual variation between liposomes cannot be excluded, indeed it would be expected within a certain extent, but, according to our data, the encapsulation of DOX was homogeneous.

Why, when considering the mechanism leading to apoptosis, were we limited to considering caspases 3, 8, and 9. Caspases 6, 7, and 10 were excluded for some reason at the preliminary stage?

We agree with the reviewer that other caspases than caspase-3, -8 and -9 could be involved in apoptosis induced by TRAIL. In this regard, although the initiator caspase-10 can be involved in TRAIL-induced apoptosis, the main initiator caspase described in the first steps of apoptotic cascade induced by TRAIL is caspase-8. In fact, it has been reported that caspase 10 cannot compensate for loss of caspase-8 despite effective recruitment to the DISC upon TRAIL triggering in the absence of caspase-8, assigning caspase-8 a central role in the initiation of extrinsic apoptosis by TRAIL (Sprick MR, et. al. EMBO J 2002; 21: 4520-3). In this line, our results (Figure 3) indicated that caspase-8 is the main initiator caspase in LUVDOX-TRAIL-induced apoptosis since apoptosis was fully abrogated when the specific caspase-8 inhibitor IETD-fmk was used. In the same way, among the executioner caspases -3, -6 and -7, caspase-3 is the main caspase executing TRAIL-induced apoptosis (Micheau O and Mérino D. Leukemia. 2004 Oct;18(10):1578-80). Finally, depending on the cell type, activation of apoptosis through the extrinsic pathway may or not may require the intrinsic pathway (Ozoren N, and El-Deiry WS. Neoplasia, 2002. 4(6): p. 551-7; Rudner J, et. al. Oncogene, 2005. 24(1): p. 130-40). In type I cells, the caspase-8 ctivated at the DISC is sufficient for activation of effector. caspases and hence for apoptosis induction. By contrast, in type II cells, it appears that the levels of active caspase-8 generated at the DISC are lower and they require further signal amplification through the intrinsic pathway. It is well known that the link between the death receptor-mediated extrinsic pathway and the mitochondrial intrinsic pathway is the BH3-only protein, Bid and that the initiator caspase in mitochondrial pathway is caspase-9. For that reason, we analyzed caspase-9 activation on order to study the role of the mitochondrial intrinsic pathway in the apoptosis induced by LUVDOX-TRAIL.

Why was doxorubicin chosen from the entire spectrum of chemotherapeutic drugs? Consider, for example, the proteasome inhibitor bortezomib or the cytostatic drug paclitaxel, since these drugs significantly enhance the induction of apoptosis by TRAIL in most resistant cell lines. Justification should be added to the introduction to the article.

We agree with the reviewer that there is a wide spectrum of drugs described that enhance TRAIL-induced apoptosis including doxorubicin. In fact, we have previously demonstrated that LUV-TRAIL could benefit from the combination with other drugs in different cancer types, mainly in combination with the cdk inhibitor flavopiridol but also with the cdk9 selective inhibitor SNS-032, the proteasome inhibitor bortezomib and the DNA-damaging agente trabectedin (De Miguel D, et. al. Recent Pat Anticancer Drug Discov. 2016;11(2):197-214; De Miguel D, et. al. Nanotechnology. 2016 May 6;27(18):185101. doi: 10.1088/0957-4484/27/18/185101; Pawlak A, et. al. Anticancer Res. 2017 Dec;37(12):6655-6665; Gallego-Lleyda A, et. al. Int J Mol Sci. 2018 May 13;19(5)). Therefore, encapsulation of any of such drugs would be a plausible experimental approach. In fact, we are currently working on generating LUV-TRAIL with flavopiridol encapsulated inside and expect to have results soon.

Concerning the drug to be encapsulated, after performing a research work and considering several options among the already-described liposomal-drugs, liposomal-doxorubicin was finally chosen. In addition to doxorubicin being reported to synergized with TRAIL (as widely referenced in the manuscript), this choice was due to liposomal-doxorubicin was one of the first encapsulated drugs to be developed, more than 20 years ago, and a lot of information about its preparation and characterization can be easily found in the literature (reviewed in: Barenholz Y. J Control Release, 2012. 160(2): p. 117-34). Moreover, from an experimental point of view, the encapsulation protocol of doxorubicin perfectly suited the coupling process of TRAIL with LUV, so both processes (encapsulation and coupling) could be easily combined with minor modifications.

Following the reviewer’s recommendations, a paragraph justifying the choice of doxorubicin has been added in the Introduction.

Why is only a 10-fold dose reduction considered? Perhaps this is not the best option. Explain your choice, please.

Indeed, after observing that LUVDOX-TRAIL were toxic on isolated peripheral blood mononuclear cells (PBMC) and on activated T-cells, we tested different concentrations of encapsulated doxorubicin inside LUVDOX-TRAIL and we found that a 10-fold lower concentration was the more appropriate dose in order to maintain its cyotoxicity against tumor cells without being toxic on normal cells.

The main problem when using anthracyclines (doxorubicin, epirubicin) is cardiotoxicity, characterized by irreversible damage to the heart muscle with a decrease in its contractile function, which can be manifested not only by a decrease in the ejection fraction of the left ventricle, but also by clinical signs of heart failure. Cardiotoxicity is cumulative and occurs when a certain total dose of the drug is reached. Liposomal doxorubicin does not reduce the effectiveness of the treatment; the toxicity spectrum remains similar, while cardiotoxicity phenomena are less pronounced. The section describing toxic effects does not mention the possible cardiotoxicity of this drug. Why?

We fully agree with the reviewer that cardiotoxicity is the main problem when using doxorubicin. Unfortunately, now we longer have organs of mice used in the experiment of in vivo toxicity, so we cannot show slides of cardiac tissue stained with hematoxylin and eosin. However we had some sera obtained from blood samples of those mice (2 mice in control group and 4 mice in LDT 1/10 group). Therefore, we have analyzed by automated immunoanalysis the cardiac enzymes creatine-kinase (CK) and creatine-kinase myocardial band (CK-MB) as serological biomarkers for cardiac damage (see Table below).

CTRL

LDT 1/10

Conc. (IU/L)

Conc. (IU/L)

p

CK

63.5±4.9

77.0±19.7

NS

CK-MB

72.5±21.9

60.5±11.1

NS

As shown in table, no statistical differences are observed between mice from control group and mice treated with LUVDOX-TRAIL 1/10. Therefore, these data indicate that LUVDOX-TRAIL do not present cardiotoxicity.

Many polyphenolic compounds, in most cases flavonoids, exhibit a synergistic effect with TRAIL, affecting various proteins involved in the regulation of apoptosis, survival or the rate of division of tumor cells. For example, quercetin can enhance the effect of TRAIL due to dephosphorylation of Akt and activation of caspases on human adenocarcinoma cells. No cytotoxicity was detected with respect to normal cells. Quercetin has been shown to be able to activate caspases-3, -8 and -9. This may be a promising area for your research. It should be added to the discussion of the results, as well as the options for using Apo2L/TRAIL in combination with other chemotherapeutic drugs.

Following the reviewer’s recommendations, we have added a paragraph discussing these possible combinations in the manuscript. Since combination strategies are indicated in the Introduction, we have considered more appropriate to add a new paragraph about combinations of TRAIL with other drugs in this section.

The therapeutic use of TRAIL is limited, since a significant portion of the tumor cells are insensitive to it. One of the reasons for this is the interaction of the cytokine with the receptor "traps" of DcRl and DcR2, which not only do not trigger the apoptotic signal, but also inhibit the signal of the death receptor. The authors obtained a unique DR5 selective mutant variant of TRAIL DR5-B, which binds to the DR5 receptor as effectively as the original cytokine, but practically does not bind to the DR4 and DcR1 receptors, and its affinity for DcR2 is reduced by 400 times (ME Gasparian, et al. Mutations enhancing selectivity of antitumor cytokine TRAIL to DR5 receptor increase its cytotoxicity against tumor cells. Biochemistry. 2015; 80 (8): 1298-1311). In this work, we performed a comparative analysis of the cytotoxic activity of wild-type TRAIL and DR5-B on 12 different tumor cell lines and two types of normal cells. In nine out of 12 lines, DR5-B killed 1.5–5 times more tumor cells compared to TRAIL and did not show toxicity to normal cells. Chemotherapy drugs, such as doxorubicin, paclitaxel, and bortezomib, enhanced the effect of the TRAIL variants, but on most cell types the enhancement effect was more pronounced for DR5-B. The half-maximum effective concentrations (EC50) for DR5-B when combined with chemotherapy were 1.5-10 times lower compared to wild-type TRAIL. Thus, the DR5-B preparation can be considered as a promising drug, both for monotherapy and for combination. The method described in the peer-reviewed article undoubtedly deserves attention, but the authors did not say anything about the limitations of the study and alternative options. Any chemotherapeutic drug will be toxic to healthy cells, including, despite dose adjustments. Consider using the mutant TRAIL DR5-B variant (or other option) in combination with your designs. Perhaps this will give the best therapeutic effect in reducing systemic toxicity.ith chemotherapy, in the treatment of TRAIL resistant tumors.

We are also entirely in agreement with the reviewer that one of the ways to reintroduce TRAIL in clinics is the use of novel variants of the death ligand with an improved bioactivity alone or in combination with other molecules that sensitize to TRAIL or act in synergy with it. We strongly believe in it and for that reason, we have been working for nine years developing novel TRAIL formulations based on its anchoring on liposome surface and also combining TRAIL with other drugs. In this line, the mutant TRAIL DR5-B exhibited a enhanced cytotoxic activity against cancer cells both alone and in combination with chemotherapic drugs. Therefore, we are delighted to collaborate with other groups working in the same line of research in order to improve TRAIL anti-tumor activity.

Finally, following the reviewer’s recommendations, we have included a new paragraph in Introduction mentioned the work carried out by ME Gasparian and co-workers with the mutant TRAIL DR5-B.

Looking forward to hearing from you. Yours sincerely,

Dr. Luis Martinez-Lostao, for the authors,

Luis Martinez-Lostao, M.D., Ph.D.

Servicio de Inmunología,

Hospital Clínico Universitario Lozano Blesa,

C/ San Juan Bosco 15,

Zaragoza,

50009 Spain.

Email: lumartin@unizar.es

Reviewer 2 Report

In this article, the authors have investigated the cytotoxic caused by liposomes carrying soluble TRAIL designated as (LUV-TRAIL), and cytotoxic caused by liposomes carrying TRIAL plus doxorubicin (DOX) in the lumen of liposomes-designated as (LUVDOX-TRAIL).

Authors concluded that, LUVDOX-TRAIL mediated in vitro cytotoxicity was much higher, and exhibited much faster kinetic as compared to LUV-TRAIL alone. The same was applied to in vivo in a tumor xenograft model. Authors also show that the action of the liposomal DOX (LUVDOX) and TRAIL (LUV-TRAIL) was specific of the liposomal DOX, and was not observed when with soluble DOX.

The study is well designed, and the conclusions are supported by experimental evidence, and data is well organized with statistics are described along with figures.

While the study holds a scope, there are some points which authors may wish to improve. My comments are appended below. 

1. Authors show that the cytotoxicity induced by liposomes carrying soluble TRAIL (LUVDOX-TRAIL) was proven to rely on two independent processes:

a dynamin-mediated internalization of the doxorubicin-loaded particle, and the activation of caspase-8 exerted by the liposomal TRAIL

How authors would distinguish whether the cell death/apoptosis was by caspase cascade activated by TRAIL, and here DOX did not have the role, or vice a versa DOX caused the cell death and TRAIL cascade was not involved. In fact in a biological system, two or more processes could go in parallel and it remains hard to provide a definitive answer whether it is nearly possible to claim the two processes are independent and are not happening simultaneously.

2. Authors have not shown the size destruction. Whether the size of particles was uniform, heterogeneous and what was the size range. It has been established that various sizes behave differently, and their biodistribution varies based on sizes, as the size effects the internalization which means some organs, cell types take one range of size while others uptake other range of size. In current case, authors have not shown the sizes, and size range and distribution.

3. Additionally, in a recent study published in ‘’Science’’, few days ago (https://advances.sciencemag.org/content/5/11/eaax9250), shares many aspects of current study reading DOX delivery by nanoparticles to targeted cells, killing specific population of cells. I would suggest authors to consider this article and extract important information from this article. Authors may also wish incorporate some changes to data presentation according to this article, which I believe will further improve the quality of paper.

4. Finally, in figures, where statistics is involved, please replace the bar charts by dot plots to show the position/distribution of individual biological replicates. Apply, where applicable. Please see Weissgerber TL, Milic NM, Winham SJ, Garovic VD. (2015) Beyond Bar and Line Graphs: Time for a New Data Presentation Paradigm. PLoS Biol 13(4): e1002128.

Author Response

Zaragoza, November 27, 2019

Cassidy Wang

Assistant Editor

Dear Editor,

Thank you very much for considering our manuscript for revision, giving us the opportunity to reply to the reviewers´ questions. Please, find enclosed a final copy of the revised version of our manuscript by De Miguel D. et al., entitled "Double-edged lipid nanoparticles combining liposome-bound TRAIL and encapsulated doxorubicin showing an extraordinary synergistic pro-apoptotic potential" (cancers-645176), and a point-by-point reply to reviewer’s suggestions. The authors would like to thank the work made by the reviewer on our paper, which have resulted in a net improvement of the initial version of the manuscript.

Nevertheless, before proceeding to discuss point by point the suggestions made by the reviewer 2, we would like to point out that given the short deadline we were given, we could only address experimentally some of the point raised by the reviewer.

Reviewer 2:

The study is well designed, and the conclusions are supported by experimental evidence, and data is well organized with statistics are described along with figures.

While the study holds a scope, there are some points which authors may wish to improve. My comments are appended below.

Authors show that the cytotoxicity induced by liposomes carrying soluble TRAIL (LUVDOX-TRAIL) was proven to rely on two independent processes: a dynamin-mediated internalization of the doxorubicin-loaded particle, and the activation of caspase-8 exerted by the liposomal TRAIL. How authors would distinguish whether the cell death/apoptosis was by caspase cascade activated by TRAIL, and here DOX did not have the role, or vice versa DOX caused the cell death and TRAIL cascade was not involved. In fact in a biological system, two or more processes could go in parallel and it remains hard to provide a definitive answer whether it is nearly possible to claim the two processes are independent and are not happening simultaneously.

We agree with the reviewer that the dissection of the individual contribution of DOX and TRAIL to the cell death induced by LDT is indeed not trivial. We will try explain the rationale we followed in order to draw the conclusions we present in the manuscript.

First, LUVDOX without TRAIL on their surface were not able to induce neither a noticeable caspase activation (Figures 3c and 4e) nor significant cell death within the first 12 hours of treatment. This allowed us to conclude that the encapsulated DOX alone was not responsible for the strong caspase activation and cell death observed upon LDT treatment. On the other hand, LT alone, without DOX, were only able to induce a limited caspase activation and cell death. In addition, it should be noted that combinations of LT with increasing concentrations of soluble DOX never reached the cell death levels of LDT. Altogether, the results suggested that the strong cytotoxicity of LDT was a unique synergistic effect, not a mere addition of the individual contributions of DOX and TRAIL, since it was significantly higher than the addition of the cytotoxicity of both TRAIL and DOX alone. On the other hand, the use of caspase inhibitors allowed us to identify caspase-8 as the main and apical factor in LDT induced cell death. In this regard, pre-incubation with the specific caspase-8 inhibitor IETD-fmk abolished LDT-induced apoptosis (Figure 3e). On the other hand, preincubation with the TRAIL blocking antibody RIK completely inhibited LDT induced cell death, even though DOX fluorescence could still be observed within the cells (as assessed by both FACS and confocal microscopy). This suggests a capital role for TRAIL in the cell death exerted by LDT, but not in the uptake of the liposomal DOX. Therefore, it could be concluded that LDT-induced cell death was dependent on caspase-8, and on TRAIL. Given the known apical role of caspase-8 in TRAIL-induced apoptosis, and considering the previous observation that both LT and LDT, but not LD, were able to induce caspase-8 activation, we concluded that the observed activation of caspase-8 was exclusively mediated by TRAIL, and not by DOX.

However, despite the capital role of TRAIL in LDT-induced cell death, only the concomitant presence of encapsulated DOX and surface-displayed TRAIL was able to induce the strong synergistic pro-apoptotic effect. In this regard, the question remained of how DOX contributed to the cell death. To answer this question, we focused on the cell internalization and DOX-liberation processes. As already mentioned, internalization of LD and LDT followed similar kinetics. This internalization was, at least partially, a dynamin-mediated process, since the dynamin inhibitor Dynasore greatly delayed the uptake process. However, when analyzed by confocal microscopy, significant differences were observed regarding DOX translocation to the nucleus (where it actually exerts its cytotoxic activity) between LD and LDT: in LDT, DOX could be found in the nucleus with a much faster kinetic than LD. Moreover, this fast translocation to the nucleus appeared caspase-dependent, since this was inhibited by the caspase inhibitor ZVAD.

As mentioned by the reviewer, there were apparently two parallel processes: a dynamin-mediated internalization of the doxorubicin-loaded particle, and the activation of caspase-8 exerted by the liposomal TRAIL. These two processes would happen individually of each other in either LD without TRAIL, or in LT without DOX (as suggested by our data: LT and LDT have a similar caspase-8 activation capacity, and LD and LDT have a similar internalization kinetics). However, when combined in a single entity, these two processes converge and result in the exceptionally strong cytotoxicity of LDT. In any case, probably, the sentence including “rely on two independent processes” could be regarded as an overstatement, and may lead to confusion. Therefore, we rewrote it in order to express more clearly our intention.

Authors have not shown the size destruction. Whether the size of particles was uniform, heterogeneous and what was the size range. It has been established that various sizes behave differently, and their biodistribution varies based on sizes, as the size effects the internalization which means some organs, cell types take one range of size while others uptake other range of size. In current case, authors have not shown the sizes, and size range and distribution.[1]

Indeed, we have not shown the size distribution of liposome population. In previous work, we determined the size of LUV (alone or coated with TRAIL) by dynamic light scattering (DLS) (De Miguel D, et. al. Mol Pharm. 2013 Mar 4;10(3):893-904). Examination of the size distribution of the vesicle suspension by DLS showed that LUV alone showed a Z-average of 155 nm while the Z-average of LUV-TRAIL was 178 nm. Therefore, binding soluble TRAIL to LUV produced little changes in the size and homogeneity of the liposome population.

On the other hand, a parameter used to define the size range and particle size distribution of the lipidic nanocarrier systems is called the Polydispersity Index (PdI). This term is used to describe the degree of non-uniformity of a size distribution of particles. In this line, PdI of liposome populations were of 0.081 for LUV alone and 0.191 for LUV-TRAIL. In drug delivery applications using lipid-based carriers, such as liposome and nanoliposome formulations, a PDI of 0.3 and below is considered to be acceptable and indicates a homogenous population of lipid vesicles (Reviewed in : Danaei M, et.al. Pharmaceutics. 2018 May 18;10(2)). Therefore, our liposome populations showed a great homogeneity both without TRAIL and with TRAIL anchored on their surface. Finally, though not quantitative, the electron microscopy analysis revealed a similar size distribution among the liposomal population. Considering these results, we assumed that the encapsulation of DOX inside LUV-TRAIL do not significantly modify homogeneity of the liposome population. Unfortunately, given the short deadline we were given (five days), we cannot only address experimentally this assumption.

Additionally, in a recent study published in ‘’Science’’, few days ago (https://advances.sciencemag.org/content/5/11/eaax9250), shares many aspects of current study reading DOX delivery by nanoparticles to targeted cells, killing specific population of cells. I would suggest authors to consider this article and extract important information from this article. Authors may also wish incorporate some changes to data presentation according to this article, which I believe will further improve the quality of paper.

We wish to thank the reviewer for drawing to our attention to the recently published work in Science Advances (Erythrocyte leveraged chemotherapy (ELeCt): Nanoparticle assembly on erythrocyte surface to combat lung metastasis. Zhao Z, et al. Science Advances, 2019, Nov 1st, 5(1):1-12). Indeed, as the reviewer mentions, this work shares some aspects with our work.

Following the reviewer’s recommendations we have incorporated a paragraph in Introduction discussing the significant results obtained in this article.

Finally, in figures, where statistics is involved, please replace the bar charts by dot plots to show the position/distribution of individual biological replicates. Apply, where applicable. Please see Weissgerber TL, Milic NM, Winham SJ, Garovic VD. (2015) Beyond Bar and Line Graphs: Time for a New Data Presentation Paradigm. PLoS Biol 13(4): e1002128.

Following the reviewer’s recommendations, we have replaced, were applicable, the bar charts by dot plots to show the distribution of individual biological replicates (Figure 6).

We hope that the revised version of our manuscript will be now acceptable for publication in Cancers.

Looking forward to hearing from you. Yours sincerely,

Dr. Luis Martinez-Lostao, for the authors,

Luis Martinez-Lostao, M.D., Ph.D.

Servicio de Inmunología,

Hospital Clínico Universitario Lozano Blesa,

C/ San Juan Bosco 15,

Zaragoza,

50009 Spain.

Email: lumartin@unizar.es